# Validation of Citizen Science Meteorological Data: Can They Be Considered a Valid Help in Weather Understanding and Community Engagement?

**DOI:** 10.3390/s24144598

**Published:** 2024-07-16

**Authors:** Nicola Loglisci, Massimo Milelli, Juri Iurato, Timoteo Galia, Antonella Galizia, Antonio Parodi

**Affiliations:** 1CIMA Research Foundation, 17100 Savona, Italy; 2IOTOPON Srl, 09126 Cagliari, Italy; 3CNR, Institute of Applied Mathematics and Information Technologies (IMATI), 16149 Genova, Italy

**Keywords:** citizen science, MeteoTracker, meteorology, I-CHANGE

## Abstract

Citizen science has emerged as a potent approach for environmental monitoring, leveraging the collective efforts of volunteers to gather data at unprecedented scales. Within the framework of the I-CHANGE project, MeteoTracker, a citizen science initiative, was employed to collect meteorological measurements. Through MeteoTracker, volunteers contributed to a comprehensive dataset, enabling insights into local weather patterns and trends. This paper presents the analysis and the results of the validation of such observations against the official Italian civil protection in situ weather network, demonstrating the effectiveness of citizen science in generating valuable environmental data. The work discusses the methodology employed, including data collection and statistical analysis techniques, i.e., time-series analysis, spatial and temporal interpolation, and correlation analysis. The overall analysis highlights the high quality and reliability of citizen-generated data as well as the strengths of the MeteoTracker platform. Furthermore, our findings underscore the potential of citizen science to augment traditional monitoring efforts, inform decision-making processes in environmental research and management, and improve the social awareness about environmental and climate issues.

## 1. Introduction

Citizen science involves the active participation of the public in scientific research, allowing individuals to contribute to data collection, analysis, and interpretation [1,2]. This collaborative approach has gained momentum in various fields, including ecology, astronomy, and environmental science [3], due to its potential to generate large datasets and engage communities in scientific inquiry [4].

In the context of environmental monitoring, citizen science offers several advantages, such as increased spatial coverage, cost-effectiveness, and enhanced public awareness of environmental issues [5,6]. Considering traditional technology-based tools and serious games, the effectiveness of citizen science initiatives is becoming a matter of fact [7,8]. When focusing on the involvement of volunteers in data collection efforts, citizen science projects can fill gaps in traditional monitoring networks and provide valuable insights into local climate patterns [9]. Furthermore, through the integration of citizen-generated observations with existing meteorological datasets, researchers could enhance the spatial and temporal resolution of climate analyses [10]. On the other hand, the most recognized barrier to the use of observations collected through citizen science monitoring initiatives relates to their perceived lack of quality, since information are derived from unofficial sources [11]; thus, it is mandatory to ensure (and document) the application of solid methodologies to verify and validate data [12,13,14].

The I-CHANGE project [15] stands as a notable example of leveraging citizen science in meteorological research. Initiated in 2021, the I-CHANGE (Citizen Actions on Climate Change and Environment) project is based on the idea that citizens and the civil society have a central role in the definition of environmental protection and climate action and that their direct involvement is essential to drive a true shift and promotion of behavioral changes toward more sustainable patterns.

I-CHANGE aims to raise awareness on the impacts of climate change and related natural hazards to enable behavioral change, building on three conceptual pillars:Improving the knowledge of climate change science and the understanding of related physical, socioeconomic and cultural processes;Promoting the active participation of citizens in data collection in eight Living Labs (LLs), in Europe, the Middle East, and Africa, through citizen tools and sensors;Increasing the usability and interoperability of the data collected by the citizens at a broader scale.

A key component of the I-CHANGE project was the establishment of a network of citizen observers equipped with in situ weather monitoring instruments and meteorological sensors on the move. Participants were provided with training and resources to set up in situ weather stations in their local communities (schools, hiking refuges), enabling them to collect data on temperature, humidity, precipitation, and other meteorological parameters. The data collected through the I-CHANGE in situ weather stations provided valuable insights into regional atmospheric patterns and trends for the wider audience represented by passionate students, teachers, hikers, and cyclists. Additionally, an on-the-move sensor was adopted: a mini-weather station MeteoTracker (hereafter MT), specifically designed and patented for measurements on the move, which was equipped with a system (RECS, Radiation Error Correction System) that allows very high measurement speed. The mini-station is “instantly” installed on the vehicle rooftop thanks to its magnetic base and on bikes and for other uses by means of a specific accessory (“bike holder”). MT was chosen for its user-friendly interface, accessibility, and ability to crowdsource meteorological measurements: by harnessing the collective efforts of volunteers, MT facilitated the collection of high-resolution weather data across diverse geographical regions, complementing traditional monitoring efforts.

In this paper, we present and discuss the validation procedure based on statistical methods we have applied to validate MT observations against the official Italian Civil Protection weather network; the overall analysis demonstrated the high quality and reliability of collected data. The paper is organized as follows: in the next section, we introduce the MT devices, providing details both on hardware technologies and the software architecture accompanying the devices; Section 2 focuses on the methodology used to collect and validate data; thus, we introduce the urban areas used in the study and the volunteers participating in the I-CHANGE initiatives and campaigns. The actual validation process, the discussion of the observations, the analysis and the results are presented in Section 4. Section 5 explains the results in terms of the effectiveness of the data and also supports the value of citizen engagement in science and in particular in climate change. Conclusions of the work are summarized in Section 6.

## 2. The MeteoTracker Device

The MT device is a low-cost, patented weather station (Figure 1) specifically designed for mobile measurements that samples several meteorological variables while moving jointly with its carrying vehicle (mobile sensor). It is based on a patented, dual-sensor, differential, negative feedback solution (RECS, Radiation Error Correction System) that deploys two identical temperature sensors (sensor 1 and sensor 2) that have different exposures to the solar radiation (different transmission coefficients in the short-wave radiation band).

The difference in radiation exposure is predefined so that the difference in the temperature reading from sensor 1 and 2 can be used to compute the error caused by solar exposure.

In addition to solar radiation error correction, the RECS is used to cancel the dependency of temperature readings vs. vehicle speed caused by the variation in the ventilation flow hitting the temperature sensor (the lower the speed, the lower the ventilation; the higher the radiation error, the higher the corrective factor generated by the RECS algorithm). This is the negative feedback mechanism that removes any dependency of the temperature measurement vs. vehicle speed.

The MT device includes temperature, relative humidity, and pressure sensors and allows for the measurement of the three related parameters plus six additional derived parameters: the vertical component of the thermal gradient, dew point, humidex, solar intensity indicator (based on the RECS calculation), potential temperature, altitude, and speed (GNSS derived).

For the temperature sensors, we used the Texas Instruments TMP117, while the relative humidity sensor was the HDC3022 sensor with a transparent membrane on top to preserve high accuracy even under saturated conditions and measurement stability in the long term.

The pressure sensor was the Bosch BMP390.

The MT device has a compact size (75 mm × 75 mm × 35 mm), and its mechanical features are the result of an engineering effort to maximize the measurement speed and ease of installation.

The battery life is in the range of 10-18 weeks (depending on environment conditions, air temperature mainly), and a small accessory solar panel is available for removing any need of recharging.

MT is available in two different solutions: MeteoTracker “smartphone” (user-activated measurement sessions by the mobile app) and “standalone” (complete automation of data collection thanks to an auxiliary cellular and GNSS modem that activates the session based on the dynamic status of the vehicle), and it is associated with a software infrastructure for real-time data transmission to the server, real-time and archive data visualization, and a set of data services.

The software infrastructure comprised the following:A mobile app (Android and iOS) for real-time data visualization and upload to the server and archive management;A full-stack suite that includes an advanced dashboard for data visualization (on map, graphs, numeric format, and other advanced features);An API service for data integration on third-party platforms;An interactive map embeddable in third-party websites;Other software implementations for data processing (like the Virtual Fixed Station suite).

## 3. Methodology

I-CHANGE is built upon the belief that the development and strengthening of citizen science initiatives to engage citizens concretely in the collection of environmental data through tools, such as new apps and friendly yet reliable devices, and the creation of a scientific-ground loop of knowledge are mandatory to improve the social awareness about environmental and climate issues and to enable behavioral and consumption shifts toward sustainable patterns. The robust citizen-engagement of I-CHANGE places local contexts, cultures, and creativity at the center of innovation to better mold the opportunities offered by new citizen-tailored innovative technological solutions and make explicit a loop of knowledge toward change impacts on the environment. To have this goal founded on solid real-life ground, the project provided a transdisciplinary knowledge synthesis of global and local challenges associated with climate change and environmental impacts. In this context, this paper describes (a part of) the experience from the Living Lab of Genoa with respect to volunteer recruitment and engagement, while the main focus is on data analysis. The methodology used in the overall work can be summarized in the following diagram (Figure 2). Volunteers were equipped with an MT weather station, which alongside the weather variables a user can monitor are described in detail in the previous paragraph as well as in [12,14,16].

Volunteers willing to use MT during their normal activities were recruited in the Genoa municipality, one of the I-CHANGE LLs, through various channels, including social media, community outreach events, and collaboration with local schools and environmental organizations [17] in agreement with the quadruple helix approach [18]. Engagement strategies, such as training workshops, educational materials, and interactive data visualization tools, were employed to motivate and retain participants [1].

The Genoa LL monitoring campaigns have focused on three different areas corresponding to the activities of three different stakeholders.

The western portion of the Genoa municipality, corresponding to the Leira catchment, has been covered by Turchino Outdoor, which is an association formed by a group of citizens with a passion for mountain biking. The Leira catchment covers an area of 27 km^2^ and has experienced flash flood episodes in 1833, 1862, 1864, 1867, 1915, 1970, 1993, and 2010, resulting in significant social and economic impacts. The Turchino gap (at 532 m), as part of the Leira catchment, plays an important role in determining local-scale effects of the “Genoa Low”, which is a cyclone that forms or intensifies from a pre-existing cyclone to the south of the Alps with an orographic effect [19] over the Ligurian Sea. Turchino Outdoor citizen scientists have extensively monitored this area since the middle of 2022 using their own mountain e-bikes equipped with MT sensors, collecting more than 150 tracks. Their activities have been also documented in newspapers and media [20].

Conversely, the Genoa urban area has been monitored by MT sensors installed on public transportation buses belonging to AMT Genoa, Azienda Mobilitá e Trasporti. AMT is a joint-stock company that holds the concession for public transport in the Italian city of Genoa and transported more than 120 million passengers in 2023. Three AMT buses covering lines along the coast as well as the Valpolcevera and Bisagno valleys have been active since the middle of 2023, collecting more than 700 tracks. The Valpolcevera and Bisagno valleys are among the most populated areas of the Genoa municipality (with about 360,000 inhabitants, constituting 45% of the total Genoa population) and have historically been affected by severe flash flood episodes (1970, 1992, 1993, 2010, 2011, and 2014) as well as by growing impacts of heatwave phenomena.

Moving to the sea, the Consorzio Servizio Marittimo del Tigullio has monitored the area from San Fruttuoso to Sestri Levante in eastern Liguria using its boats. The Tigullio Maritime Service Consortium provides tourist transport throughout the Gulf of Tigullio, with year-round connections from Rapallo and Santa Margherita Ligure to Portofino and San Fruttuoso, and seasonal departures from Chiavari, Lavagna, Sestri Levante, and Moneglia, especially on weekends and during periods of greater affluence. Until the end of 2023, they were collecting 750 tracks. This area of Liguria is affected by extreme hydro-meteorological events [21,22], and the availability of MT sensors on these boats is enabling a deeper understanding of complex phenomena at the interface between the sea and atmosphere, as well as sea–land thermodynamics, in close cooperation with citizens.

## 4. Results

Statistical methods, including time-series analysis, spatial and temporal interpolation, and correlation analysis, were employed to analyze the collected data and verify them in order to identify patterns, trends, and relationships among the meteorological variables.

The analysis of MT data yielded a comprehensive dataset comprising meteorological observations from multiple locations and time periods.

The reliability and trustworthiness of low-cost sensors, in fact, are typically at stake when used outdoors as large and sharp transitions in the environmental conditions (sudden temperature drops, wind gusts, etc.), as well as extreme regimes (saturation, rain, snowfall, etc.), are often demanding for sensor accuracy [16].

The consistency and accuracy of citizen-generated data were evaluated by comparing MT measurements with data from established in situ weather stations [23]. In particular, we focused on the analysis of the temperature values retrieved from MT and those coming from the official Italian civil protection weather network, as shown in Figure 3.

Differently from most of the instrumentation designed for research purposes, the MT operates on the move, introducing a further degree of complexity to the validation procedure [16]. To counteract this limitation, the paths of the citizen science network of the I-CHANGE project coming from the Genoa LL were selected in such a way that these paths passed close to the automatic measurement stations of the official weather network (hereafter OWN). The geographical selection parameter was the distance of the MT sampling points from these stations, which was established as 200 or 300 m, depending on the size of the statistical sample, as far as measurements were on the land, while 3000 m was adopted for the measurements on the sea. The selected data were further processed to correct any difference in altitude between the MT measurement and the reference measurement of the OWN, using a wet adiabatic lapse rate adjustment.

To verify the reliability of MT thermal data, scatter plots were generated by plotting the temperature readings from the sensor of the OWN on the x-axis and the corresponding measurements on-board MT on the y-axis Figure 4 and Figure 5. Each data point on the scatter plot represents a simultaneous measurement of temperature from both thermometers.

Our initial analysis focused on assessing the agreement between the two sets of temperature data. A visual inspection of the scatter plots reveals the distribution of data points along the bisector, indicating a strong correlation between the measurements obtained from the two sources of data.

To quantify the agreement further, box plots displaying the distribution of temperature difference values between temperatures from the OWN stations and those from MT is shown in Figure 6, and statistical measures such as the root mean square error and *p*-value were calculated and are listed in Table 1. These measures provide insights into the degree of linear relationship and the overall accuracy of the temperature readings between the two sources.

The root mean square error and *p*-values were within acceptable limits, indicating consistent agreement between the two datasets.

## 5. Discussion

Overall, our comparative analysis demonstrates the reliability and concordance of temperature data collected from citizen science (MT) and conventional sources (OWN), supporting their interchangeability in temperature monitoring applications and highlighting the complementary role of citizen science in augmenting existing monitoring networks.

A more in-depth analysis of the data collected by the Consorzio Servizio Marittimo del Tigullio shows a tendency to underestimate temperature by MT with respect to the OWN. It should be underlined that the first data are recorded on board boats passing, on the sea, at most 3000 m from the monitoring points selected by the OWN, on the land. The underestimation can be attributed to the higher thermal capacity of the sea with respect to the temperature on the land, fueling the typical sea breeze.

Despite potential sources of error, such as instrument calibration, measurement bias, and data-reporting inconsistencies, citizen-generated data demonstrated high quality and reliability when subjected to rigorous quality control procedures [13,24,25]. The robustness of citizen science data was attributed to the collective efforts of volunteers, coupled with advancements in sensor technology and data validation techniques. MT exhibited several strengths, including its user-friendly interface, scalability, and ability to engage diverse stakeholders in environmental monitoring activities [26]. However, challenges, such as data privacy concerns, volunteer turnover, and data quality assurance, were identified and addressed through continuous improvement and stakeholder engagement efforts [27].

The insights gained from the analysis of MT data have important implications for environmental research, policy development, and public engagement initiatives [28,29]. Firstly, the participatory nature of the initiative, the idea of contributing to a European research project with valuable data, and the active role in each campaign fostered a sense of ownership and empowerment among citizens, encouraging continued engagement and collaboration [9,30]. Furthermore, our results underscore that citizen science can contribute to a more comprehensive understanding of climate change impacts, facilitate adaptive management strategies, and empower communities to participate in decision-making processes related to environmental conservation and sustainability [31]. To maximize the effectiveness of citizen science initiatives in environmental monitoring, stakeholders are encouraged to invest in capacity building, technology infrastructure, and data interoperability standards [28]. This is a requirement that the European Commission research programs are strongly encouraging. With these goals in mind, the I-CHANGE project is developing the so-called Environmental Impact Hub (EIH), a central component aimed at effective data sharing, management, and processing, while ensuring the usability and interoperability of heterogeneous data at European and global scales. The EIH serves observational data from the I-CHANGE Living Labs that has been ingested, standardized, quality assessed and stored. Data can be accessed through API (to facilitate developers) or through tools to support citizen engagement initiatives. The support of high-quality data, accomplished with standard interfaces for data sharing, enables the EIH to interoperate with existing and future data infrastructure at the European and global level, such as GEOSS, Copernicus, and the European Data Space [32].

## 6. Conclusions

Citizen science represents a paradigm shift in how scientific research is conducted, democratizing access to data and expertise while fostering collaboration and innovation across disciplinary boundaries [33]. With this goal in mind, the I-CHANGE project organized several participatory initiatives within the eight Living Labs; citizen and stakeholders have been engaged to participate in science through observational campaigns, dissemination events, forums, etc. Among others, MeteoTrackers as a low-cost mobile device were exploited in the observation collection; this paper discusses the validity of such citizen-based data obtained through statistical methods (time-series analysis, spatial and temporal interpolation, and correlation analysis) against measurements of the official weather network. The analysis of MT data within the I-CHANGE project has demonstrated the actual value of collected data and, as a consequence, of citizen science in enhancing the observational capacity of environmental monitoring networks. We also recognized limitations and challenges, such as the intrinsic necessity to apply data quality assurance and address data privacy concerns. Also, the volunteer turnover may impact citizen observational campaigns. However, these points are being successfully managed, and we strongly support the value of citizen-generated data. By engaging volunteers in data collection and analysis efforts, citizen science initiatives can generate valuable insights into weather patterns and climate trends at local and regional scales. The success of citizen science projects, such as I-CHANGE, underscores the potential of participatory approaches to address complex environmental challenges and promote sustainable development.

Moving forward, there is a need to further integrate citizen science into mainstream scientific practices, policy frameworks, and educational curricula [34]. By leveraging advances in technology, data analytics, and community engagement strategies, citizen science has the potential to revolutionize environmental monitoring and empower individuals to become active stewards of their local environments [35,36].

## Figures and Tables

**Figure 1 sensors-24-04598-f001:**
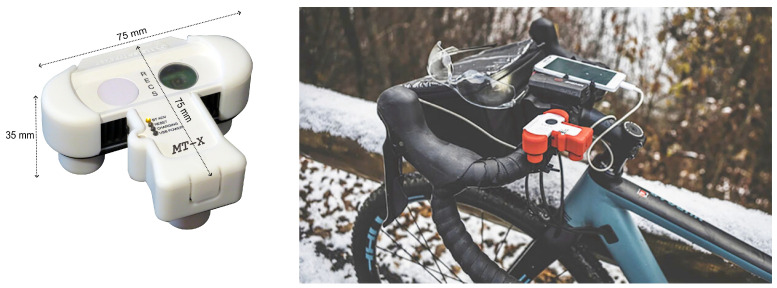
The MeteoTracker device (on the **left**) and an example of installation on board a bicycle (on the **right**).

**Figure 2 sensors-24-04598-f002:**
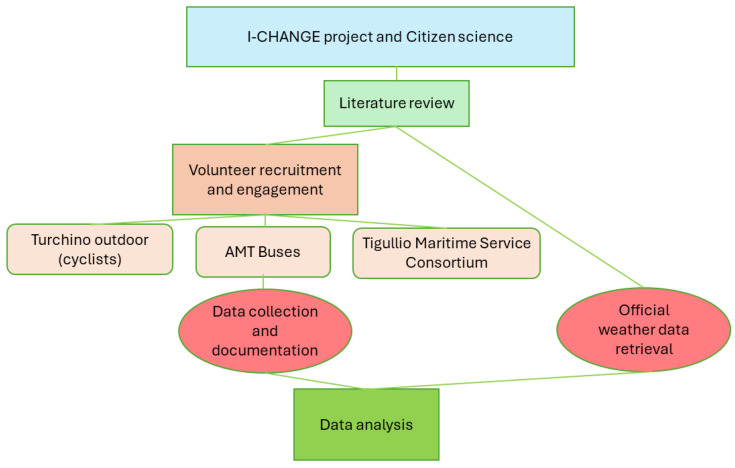
The design adopted for the research.

**Figure 3 sensors-24-04598-f003:**
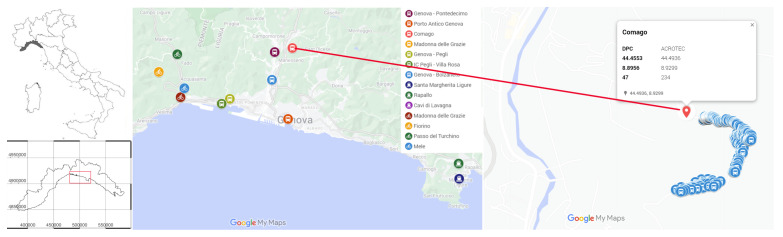
The Italian OWN surrounding the city of Genoa used for comparison (**left**) and an example of the MT points used for the validation (**right**).

**Figure 4 sensors-24-04598-f004:**
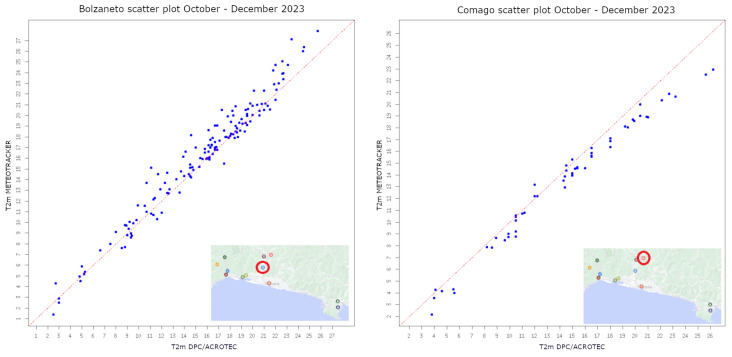
Scatter plot for temperature data coming from MT on board AMT vs. OWN stations (red circle indicates the location of the corresponding OWN station).

**Figure 5 sensors-24-04598-f005:**
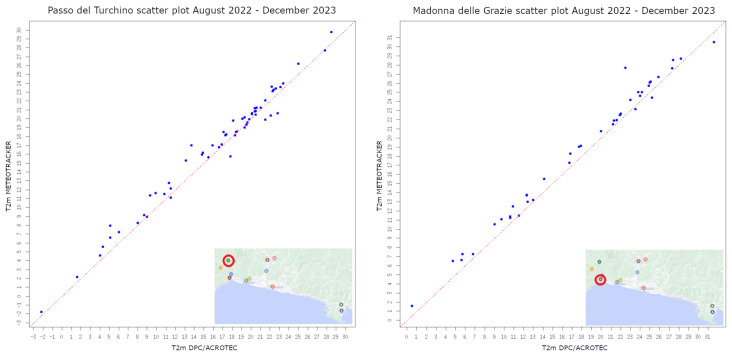
Scatter plot for temperature data coming from MT on-board bicycles and boats vs. OWN stations (red circle indicates the location of the corresponding OWN station).

**Figure 6 sensors-24-04598-f006:**
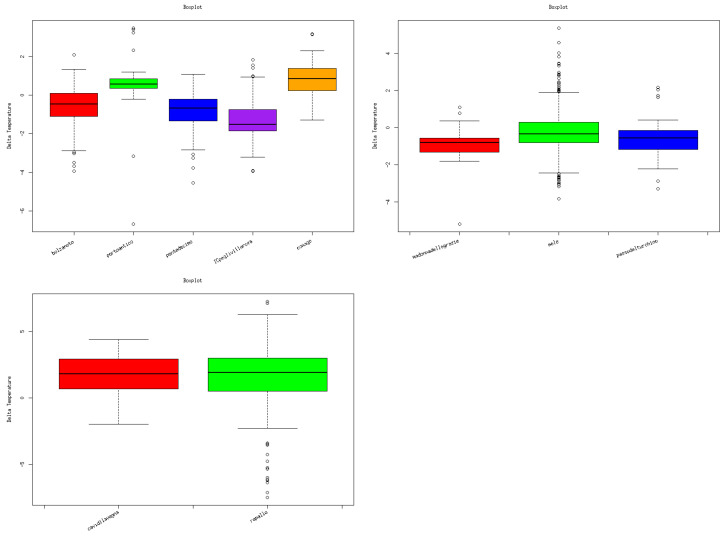
Box plots for temperature data coming from MT (AMT top left, cyclists top right, and boats bottom left) and OWN stations. With a standard notation for the box plots, the circles in the Figure represent the outlier.

**Table 1 sensors-24-04598-t001:** Statistical parameter for MT thermal data validation against OWN data.

Location	R^2^	*p*-Value
Bolzaneto	0.9624	2.2 × 10^−16^
Comago	0.9813	2.2 × 10^−16^
IC Pegli Villarosa	0.9232	2.2 × 10^−16^
Pontedecimo	0.9723	2.2 × 10^−16^
Porto Antico	0.7696	2.2 × 10^−16^
Passo del Turchino	0.9859	2.2 × 10^−16^
Madonna delle Grazie	0.9094	2.2 × 10^−16^
Mele	0.9702	2.2 × 10^−16^
Rapallo	0.7618	2.2 × 10^−16^
Cavi di Lavagna	0.8168	2.2 × 10^−16^

## Data Availability

Data from the MeteoTrackers are available for visualization on the customized platform at https://app.meteotracker.com/ (accessed on 9 July 2024) under the author name “genova_living_lab_⁎”, where ⁎ is a one-to-two digit number representing the MT ID in the location as well as under the author name “CIMA I-Change”. Reference data can be made available upon request.

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
