# Peer review of "Validation of Citizen Science Meteorological Data: Can They Be Considered a Valid Help in Weather Understanding and Community Engagement?"

_sensors, 2024, doi:10.3390/s24144598_

Round 1

Reviewer 1 Report

Comments and Suggestions for Authors

From the title and abstract it seems that the paper discusses the potential of citizen science to “enhance weather understanding” and “community engagement” but actually it focuses on the analysis of data consistency of citizen-generated data on temperature from a MeteoTracker device in Genoa municipality. It tests three different types of stakeholder data: an association of mountain biking; public transportation buses; and the Consorzio Servizio Marittimo del Tigullio although there is no clear results or further discussion on the differences between them, except for some comments on the third case. All the issues raised in the introduction, discussion and conclusion are really misleading the real contribution of the paper which is the temperature data quality comparison. 

Since the real contribution of this work is the comparison of data consistency I would recommend authors to state it clear in the title, abstract, paper’s objective and conclusion. None of this is properly formulated. Conclusions are not derived from results and should be focused on the data analysis and results that I recommend should be expanded.

Other comments

I would suggest improving data quality presentation such as figure 2 (for example, there is no geographical reference on the map’s track), readers cannot compare the location of the official data with the citizen-generated data. I also recommend developing further the data analysis and explanations.

Reviewer 2 Report

Comments and Suggestions for Authors

1.       Some systems similar with I-CHANGE aims to raise awareness on the impacts of climate change have been promoted in different countries. Some further literature review like follows might benefit your introduction section.

l  ICT-mixed community participation model for development planning in a vulnerable sandbank community: Case study of the Eco Shezi Island Plan in Taipei City, Taiwan, International Journal of Disaster Risk Reduction, 58: 102218. https://doi.org/10.1016/j.ijdrr.2021.102218

l  The levee dilemma game: A game experiment on flood management decision-making. International Journal of Disaster Risk Reduction 90: 103662. https://doi.org/10.1016/j.ijdrr.2023.103662

2.       Why MeteoTracker device? Is this device the best choice for this study? Is there evidence for that based on previous literature? What is the major limitation or short comes of it?

3.       Please show a diagram of your research design and research flow. Also, the details and a summery of your data collection should be also addressed in your 3. Methodology section.

4.       It is very weird that there is no any paragraph in your section 4.1. Figures, Tables and Schemes. Please add some explanation about these results.

5.       Please do not forget to state the major limitation of the analysis or the findings of your research in the discussion or conclusion section.

Comments on the Quality of English Language

 Extensive editing of English language required.

Round 2

Reviewer 1 Report

Comments and Suggestions for Authors

While the focus on the paper’s real contribution have improved, I still find that the presentation of results can be improved in the sense I suggested before.

Author Response

Comment: While the focus on the paper’s real contribution have improved, I still find that the presentation of results can be improved in the sense I suggested before.

Answer: We sincerely thank the reviewer for their valuable comment. In response to their suggestion, we have made modifications to the figures 4 and 5.

Reviewer 2 Report

Comments and Suggestions for Authors

Thanks for the authors for their efforts on revising the paper, I think the revised version looks acceptable for Sensors now comparing to the first version.

Comments on the Quality of English Language

Minor editing of English language required

Author Response

We sincerely thank the reviewer for their valuable comments.